# Repeated automatic sleep scoring based on ear-EEG is a valuable alternative to manually scored polysomnography

**Troels Wesenberg Kjaer**[1], **Mike Lind Rank**[2], **Martin Christian Hemmsen**[2], **Preben Kidmose**[3], **Kaare Mikkelsen**[3]*

**1** Visiting professor, Department of Neuroscience, University of Copenhagen, Denmark, **2** T&W Engineering, Lynge, Denmark, **3** Department of Electrical and Computer Engineering, University of Aarhus, Denmark

* mikkelsen.kaare@ece.au.dk

**Data Availability Statement:** A condensed version of the data set, consisting of the extracted feature values as well as labels, are available in an external

## Abstract

While polysomnography (PSG) is the gold standard to quantify sleep, modern technology allows for new alternatives. PSG is obtrusive, affects the sleep it is set out to measure and requires technical assistance for mounting. A number of less obtrusive solutions based on alternative methods have been introduced, but few have been clinically validated. Here we validate one of these solutions, the ear-EEG method, against concurrently recorded PSG in twenty healthy subjects each measured for four nights. Two trained technicians scored the 80 nights of PSG independently, while an automatic algorithm scored the ear-EEG. The sleep stages and eight sleep metrics (Total Sleep Time (TST), Sleep Onset Latency, Sleep Efficiency, Wake After Sleep Onset, REM latency, REM fraction of TST, N2 fraction of TST, and N3 fraction of TST) were used in the further analysis. We found the sleep metrics: Total Sleep Time, Sleep Onset Latency, Sleep Efficiency, Wake After Sleep Onset were estimated with high accuracy and precision between automatic sleep scoring and manual sleep scoring. However, the REM latency and REM fraction of sleep showed high accuracy but low precision. Further, the automatic sleep scoring systematically overestimated the N2 fraction of sleep and slightly underestimated the N3 fraction of sleep. We demonstrate that sleep metrics estimated from automatic sleep scoring based on repeated ear-EEG in some cases are more reliably estimated with repeated nights of automatically scored ear-EEG than with a single night of manually scored PSG. Thus, given the obtrusiveness and cost of PSG, ear-EEG seems to be a useful alternative for sleep staging for the single night recording and an advantageous choice for several nights of sleep monitoring.

## Author summary

Sleep is important to our overall health, and as such, it is valuable in a clinical context to monitor sleep, for instance in terms of quality and duration. However, the established method for performing sleep measurements, the polysomnography, is uncomfortable to wear and expensive to use. In this study we have looked at the tradeoffs involved in using a smaller, less intrusive recording method, the ear-EEG, which has slightly worse data

data repository: Mikkelsen, Kaare (2022), "Ear-EEG Sleep Monitoring Feature data ", Mendeley Data, V1, doi: 10.17632/dj6ph9k53p.1, https://data.mendeley.com/datasets/dj6ph9k53p.

**Funding:** This work was sponsored by the Innovation Fund Denmark, grant 7050-00007, which supported PK, KBM. The funders had no role in study design, data collection and analysis, decision to publish, or preparation of the manuscript.

**Competing interests:** The authors have declared that no competing interests exist.

quality for a single night, but which is more feasible to use for multiple nights on the same patient. We find that while the single-night sleep information is slightly less reliable for the ear-EEG than for the polysomnogram, this can be counter acted by simply recording multiple nights; and ear-EEG used for two nights is generally more reliable than PSG used for 1 night. Considering that multiple nights of ear-EEG may in the future be much cheaper and easier to record than a single night of polysomnography, we conclude that ear-EEG sleep monitoring in the patients own home is promising method for clinical sleep monitoring.

## Introduction

Sleep and the quality of sleep plays an essential role in health, general well-being and quality of life. Abnormal sleep is associated with mental disorders [1], and can also be an early indicator of a number of diseases in medicine and neurology e.g. Alzheimer's disease [2] and cardiovascular disease [3] and sleep monitoring may be relevant when evaluating cancer treatment [4].

During sleep, our brain alternates between sleep stages, which are characterized by specific patterns of brain and body activity. The sequence of sleep stages over a night of sleep is visualized in a hypnogram that reflects the sleep architecture. The hypnogram is fundamental to sleep medicine, because the sleep architecture, combined with other variables, provides the basis for diagnosing most sleep related disorders [5].

The gold standard for sleep assessment is manual scoring of polysomnography (PSG) recordings based on the guidelines set out in the American Academy of Sleep Medicine's (AASM) manual for the scoring of sleep and associated events [6]. The hypnogram as well as a number of other sleep metrics, such as the sleep latency, the sleep efficiency and the percentage of total sleep time in each stage can be derived from the PSG. In its full form, PSG registers electroencephalogram (EEG), electro-oculogram (EOG), electromyogram (EMG), electrocardiogram (ECG), pulse-oximetry, airflow and respiratory effort. However, when performing sleep staging in persons without respiratory disorder or respiratory issues the EEG, EOG, and EMG are often sufficient.

When we in a clinical or research context base an analysis on a single (or a few) night's sleep, it is with the implicit assumption that the individual night is a representative sample from the individual patient or subject. Every night from a person will be different and therefore two samples will not give the same result. If the person sleeps similarly from night to night, there will be little variation between the individual samples, whereas if the person sleeps differently from night to night there will be large variation between the samples. Regardless of whether there is a large or a small variance, each individual sample is presumably an unbiased estimate of the mean.

In general, if independent samples are extracted from a distribution, then the variance of the mean will decrease inversely proportional to the number of samples. Therefore, it is generally preferable to take the mean of a number of samples in order to obtain a more accurate and precise estimate.

PSG is expensive and obtrusive, and therefore not suited for more than a single or a few nights of registration. Fortunately, there are a diversity of different sleep trackers that are less expensive, less obtrusive, based on a range of different types of sensors and allows for sleep monitoring over extended periods [7–10]. These devices have in general a larger variance in the estimates of the sleep metrics than PSG. However, they may still provide valuable information [11]. Therefore, we have decided to investigate a combination based on the better of the

two methods: PSG and wearable, more precisely the ear-EEG [12]. The ear-EEG is a promising mobile sleep monitor, which has proven to be both reliable and accurate [13].

If the estimate of a given sleep metric is unbiased, the variance can be reduced by averaging over several nights. It is thus clear that there will be a balance point above which N measurements from a wearable device will give a more precise estimate than M measurements from a PSG (N> M). In this study, we have investigated the balance point between PSG and a wearable sleep-monitoring device based on ear-EEG [8,14]. More specifically, we have tested the hypothesis that ear-EEG is a useful tool for sleep scoring, and the hypothesis that automatic sleep scoring based on repeated ear-EEG is superior to a single night (M = 1) of manually scored PSG.

The promise of this approach is that given a good mobile sleep monitor, multiple wearable nights are likely to be cheaper and more practical to obtain than a single PSG night. Thus, our aim is to quantify at what amount of data wearable sleep monitoring (in this case ear-EEG) is superior to single PSG nights.

## Methods

Twenty healthy volunteers were included after verbal and written informed consent and equipped with simultaneously recording PSG and ear-EEG equipment for four nights each (Fig 1). There were 13 females, age 22 to 35, 24.9 +- 3.8 years (mean, s.d.) (Table 1). The participants were recruited primarily among the healthy student population through word of mouth. Exclusion criteria were: known sleep issues, contact-allergies to the materials, risk of sleep apnea, psychiatric disorders, sleep affecting medications and pregnancy. All recordings took place in the subject's own home. Both PSG and ear-EEG setups were produced in-house, and electrodes we connected to a biosignal recorder (Mobita, TMSi, Netherlands). The first account of the data and analysis by only a single technician has previously been presented by Mikkelsen et al [14] where a detailed account of the recordings is available, including measures of data quality. Please also see this for a detailed description of the production of the equipment.

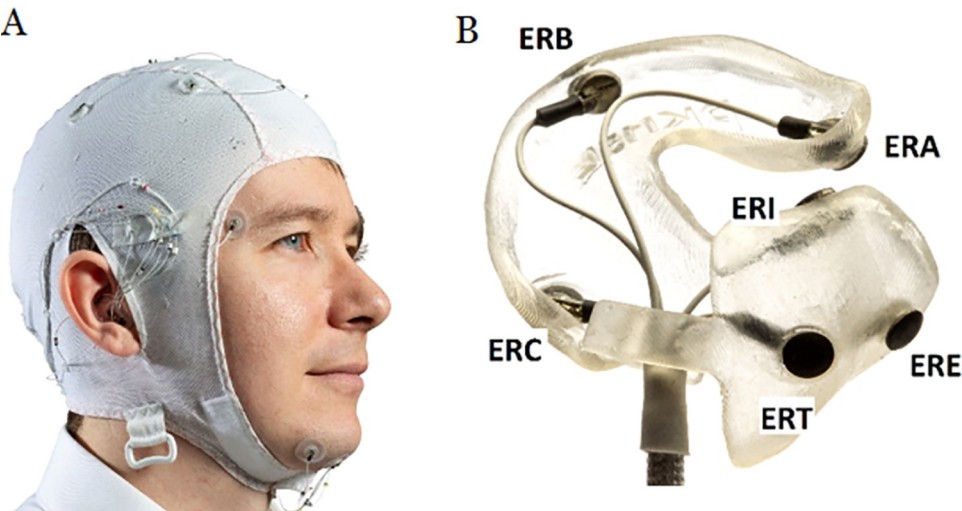

**Fig 1. The experimental setup.** A) Fully mounted participant with EOG, EMG, EEG-cap and bilateral ear-EEG. B) Right earplug with six electrodes.

**Table 1. Characteristics of study population.** Each participant is given one line. Fourteen participants had no prior experience with ear-EEG. Everyone was recorded for four nights. In 3 nights, the participant removed one earplug. The rest of those nights was not scored.

| Participant | Sex | Age | Prior ear-EEG experience | # of nights recorded (max 4) | #earplugs out before start (max 8) | #earplugs out after start (max 8) |
|---|---|---|---|---|---|---|
| 1 | F | 24 | No | 4 | 0 | 0 |
| 2 | F | 22 | No | 4 | 0 | 0 |
| 3 | F | 23 | No | 4 | 0 | 0 |
| 4 | F | 22 | No | 4 | 2 | 0 |
| 5 | F | 23 | No | 4 | 0 | 0 |
| 6 | F | 24 | No | 4 | 0 | 1 |
| 7 | F | 25 | Yes | 4 | 2 | 1 |
| 8 | F | 26 | Yes | 4 | 0 | 0 |
| 9 | M | 29 | No | 4 | 0 | 0 |
| 10 | F | 26 | Yes | 4 | 3 | 0 |
| 11 | M | 35 | Yes | 4 | 4 | 0 |
| 12 | F | 33 | Yes | 4 | 0 | 0 |
| 13 | M | 28 | Yes | 4 | 0 | 0 |
| 14 | M | 22 | No | 4 | 1 | 0 |
| 15 | M | 21 | No | 4 | 0 | 0 |
| 16 | M | 21 | No | 4 | 0 | 0 |
| 17 | F | 22 | No | 4 | 0 | 0 |
| 18 | F | 23 | No | 4 | 0 | 0 |
| 19 | F | 22 | No | 4 | 2 | 1 |
| 20 | M | 27 | No | 4 | 0 | 0 |

The study was approved by the Danish Central Region Committee on Biomedical Research Ethics (1-10-72-413-17) and the Danish Medicines Agency (2017111085).

## Sleep staging approaches

Sleep stages were classified either manually or automatically. Two certified sleep technologists (denoted S1 and S2) performed the manual scoring. The two sleep technologists did not train together, and were affiliated with different sleep clinics.

A computer algorithm did automatic scoring. The algorithm was trained on different sets of EEG channels and different manual labels, resulting in four different automatic classifiers.

We analyzed full night data in 30-second sequential epochs starting from the "light out" marking. All methods assigned one of the six classes (unclassified, W, REM, N1, N2, N3) to each epoch.

The manual scoring was based on six channels of EEG (F4-M1, C4-M1, O2-M1, F3-M2, C3-M2, O1-M2, two channels EMG (chin1-chin3, chin2-chin3) and two channels EOG (E1-M2, E2-M2) according to the AASM standard [6]. The order of the 80 nights of PSG was shuffled and anonymized before manual scoring. The two scorers are highly experienced, they trained and work at different hospitals and did not interact during the scoring period.

One classifier was trained based on the PSG data and labels from the manual scoring. The three other classifiers were based on ear-EEG from both ears and labels from the manual scoring. No other measures or priors were used for the classifiers. The algorithm consisted of a 'random forest' ensemble [15] of 100 bagged trees, fed with 77 handcrafted features extracted from 7 epochs (4 before, 2 after, the epoch being classified) from different EEG derivations. For the PSG-based classifier, the derivations were: F4-M1, C4-M1, O2-M1, F3-M2, C3-M2, O1-M2, chin1-chin3, chin2-chin3, chin1-chin2 and E1-E2, while for the ear-EEG based

classifiers, they were "between the ears" and "concha electrodes vs. ear canal electrodes" in either ear (three ear-EEG derivations in total). In the latter case, epochs for which one or both earplugs were missing were discarded. Each classifier was trained in a leave-one-subject-out fashion, meaning a 20-fold cross validation. The algorithm was implemented in MATLAB and identical to the one presented in Mikkelsen et al. 2019 [14].

## Comparison of labelling methods

We use several different sources of sleep labeling, using the following naming convention:

| | |
|---|---|
| **S1** | Manual scoring of PSG recording by sleep technician 1 |
| **S2** | Manual scoring of PSG recording by sleep technician 2 |
| **aut_psg_cons** | Automatic scoring using EMG, EOG and EEG, and trained on the consensus labels from S1 and S2 |
| **aut_ear_1** | Automatic scoring using ear-EEG and trained using the labels from S1 (manual scoring of PSG recording by scorer 1) |
| **aut_ear_2** | Automatic scoring using ear-EEG and trained using the labels from S2 (manual scoring of PSG recording by scorer 2) |
| **aut_ear_cons** | Automatic scoring using ear-EEG and trained using the consensus labels from the manual scoring of the PSG |

The obtained hypnograms (see example in Fig 2) were compared on an epoch-by-epoch basis. The nine comparisons considered were:

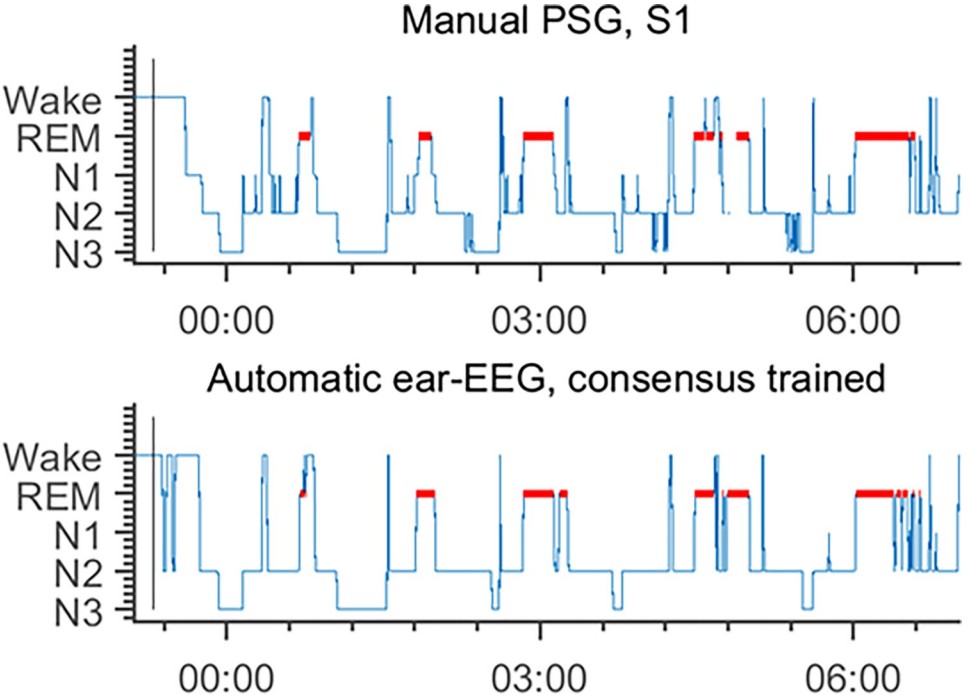

**Fig 2. An example of two hypnograms based on two different ways to do sleep scoring.** Top panel: Gold standard PSG based on six channel EEG, two channel EMG and two channels EOG. Bottom panel: Automatic ear-EEG based on 12 ear-EEG electrodes placed 6 in each ear. 30-sec epochs were considered from light out until last wake up followed by unmounting the devices. In this case, lights were out at 22:13. The Cohens Kappa value for this set of scorings is $\kappa = 0.78$. This indicates a substantial agreement and is generally acceptable. When looking more carefully the automatic ear-EEG-based scoring seems to miss a few of the nightly awakenings.

| 1 | S1 vs S2 |
| 2,3 | S1 and S2 respectively vs aut_psg_cons |
| 4,5 | S1 and S2 respectively vs aut_ear_1 |
| 6,7 | S1 and S2 respectively vs aut_ear_2 |
| 8,9 | S1 and S2 respectively vs aut_ear_cons |

For each comparison, we calculated Cohen's kappa ($\kappa$) [16]. $\kappa$ measures the agreement between the two scorers who each classified six classes (W, REM, N1, N2, N3, and unclassifiable). We also constructed confusion matrices to identify where comparisons succeeded and failed.

## Sleep metrics

Eight sleep metrics (Total Sleep Time, Sleep Onset Latency, Sleep efficiency, Wake After Sleep Onset, REM latency, REM fraction of sleep, N2 fraction of sleep, N3 fraction of sleep) were calculated for the further comparison of models. The sleep metrics were calculated as:

**Total Sleep Time**: Time spent asleep

**Sleep Onset Latency**: Time from "light out" to first sleep epoch

**Sleep efficiency:** Total sleep time divided by time from light out to light on

**Wake After Sleep Onset**: Amount of wake time, in the interval between first and last sleep epoch

**REM latency**: Time from sleep onset to first REM stage epoch

**REM percentage of sleep**: Number of epochs in REM / number of sleep epochs in (REM, N1, N2, N3) * 100%

**N2 percentage of sleep**: Number of epochs in N2 / number of epochs in (REM, N1, N2, N3) * 100%

**N3 percentage of sleep**: Number of epochs in N3 / number of epochs in (REM, N1, N2, N3) * 100%

N1 percentage was excluded, both to keep the number of comparisons down, but also because N1 scoring is notoriously unreliable, for both machines and people, making such a comparison less informative.

## Performance measures

Scatter plots were used for pairwise comparison of the labelling methods. The plots in the results section show both the identity line as well as the 'bias lines' (identity + bias). In this paper 'bias' is the average difference between technician S1 and the alternative scoring, while 's.d.' is the standard deviation of the same differences.

To be able to compare results for different sleep metrics, we also discuss 'normalized bias' and 'normalized s.d.'. These are calculated by using the 'bias' and 's.d.' numbers from above, scaled by the standard deviation of the sleep metric as based on the S1 scoring. In the case of 'bias', we have used the absolute value of the 'bias'.

To evaluate the potential benefits of averaging sleep metrics from multiple nights, we focus on the uncertainty of the within-subject estimate for each sleep metric (also known as the 'standard error of the mean'). For M nights (M from 1 to 4), the uncertainty on the within-subject mean can be calculated as:

$$uncertainty\ for\ j\ nights = \frac{1}{20} \sum_{n=1}^{20} \sqrt{\frac{\sigma_n^2}{M}}$$

Where $\sigma_n^2$ is the variance of the sleep metric from all four recordings for subject $n$. This approach is equivalent to randomly sampling j nights from each subject, calculating the average for each subsampling, and estimating the standard deviation from these averages.

For comparison, in the figures we have also included the 'population spread', which is the standard deviation of the per-subject average. This is meant to give a scale of reference for determining whether a given subject is an outlier.

This method of comparison makes it possible to compare data sets of different sizes, and to evaluate the benefits of having additional recordings.

To be clear, the necessary assumption, which must be made when a single night of PSG is used to evaluate a patient, is that the single night can be used to estimate average values of the sleep metrics for that given patient. Under that assumption, multiple recordings from the same patient when combined give better and more reliable estimates of the same average values. The possible improvement in reliability, from using more data from a less accurate method, is what we seek to investigate here.

## Results

The scorer-t0-scorer comparison of PSG-based sleep scoring when all nights and subjects are included without considering repeated measures showed a Cohen's kappa of $\kappa = 0.83$ (Fig 3).

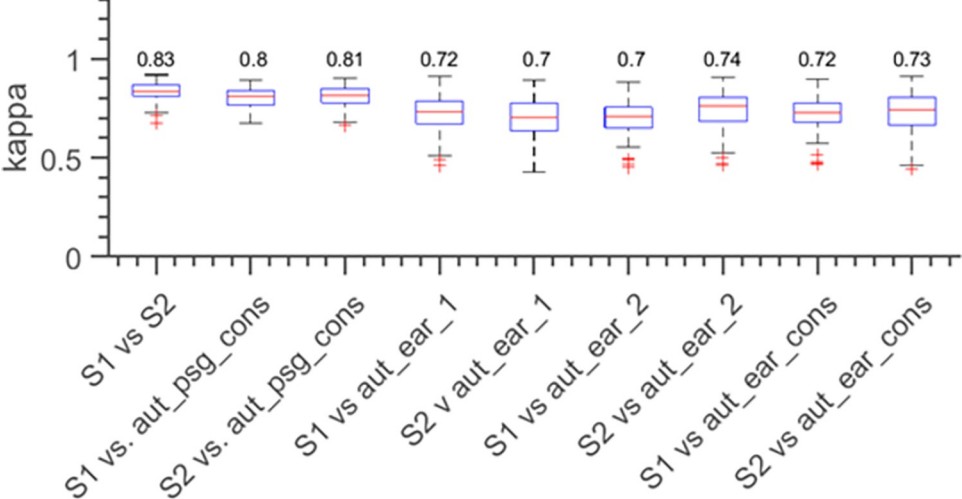

**Fig 3. Box-plots showing the agreement (Cohen's kappa) between different scorers and classifiers.** The first column 'S1 vs S2' shows the statistics for scorer1 vs scorer2. The second and third column 'S1 vs aut_psg_cons' and 'S2 vs aut_psg_cons' shows the statistics for scorer1 and scorer2 vs the automatic PSG-based sleep scoring trained on the consensus epochs. The fourth to the seventh column shows the statistics for a manual PSG-based scoring vs an automatic ear-EEG based scoring; e.g. 'S1 vs aut_ear_1' is the statistics for scorer1 vs the scoring from the automatic ear-EEG sleep scoring algorithm trained on labels from scorer1. The last two columns show the statistics for scorer1 and scorer2 vs an automatic ear-EEG based scoring trained on consensus epochs.

**Fig 4. Confusion matrices for three comparisons of raters/methods over all nights.** The classes are W(Wake), R (REM-sleep), N1 (N1-sleep), N2 (N2-sleep), N3 (N3-sleep), '?' (scorer could not classify). The numbers indicate how many 30-sec epochs were classified with this combination of scores. The '%' indicates the percentage of epochs relative to the total number of epochs. Numbers in the orange diagonal indicate agreement. In the left matrix, first label is scorer1 based on PSG and second label is scorer2 based on PSG. In the middle matrix, first label is scorer1 based on PSG and second label is an automatic algorithm based on consensus of the two scorers. The right matrix is similar to the middle, just with scorer2 instead of scorer1.

This is considered almost perfect agreement between the two scorers [17]. When the scorers were compared to automatic scoring of PSG data the agreement was a little weaker (K = 0.81). When the manual scores were compared to automatic sleep, scoring based on ear-EEG the performance was down to $\kappa$ in the range 0.7 to 0.74, indicating a substantial agreement.

In order to get a better understanding of which classes contributed the most to agreement and disagreement, we inspected the confusion matrices (Fig 4, see figure caption for explanation). The two PSG-scorers agreed in 86% of epochs, while each PSG-scorer agreed with the automatic model in only 80% and 81% of epochs respectively (Fig 4).

Night-by-night comparison of the two types of sleep scoring revealed good agreement with a median Cohens kappa value of $\kappa$ = 0.73, mean 0.72+-0.09 s.d. A single night is shown in Fig 2 for illustration.

We have looked at eight sleep metrics that are typically estimated in clinical practice. The estimate by manual PSG consensus versus automatic ear-EEG scoring demonstrates a moderate-high overlap for each metric (Fig 5). Here we see that there is good performance with high accuracy and precision between scorers and automatic sleep scoring (low bias and low s.d.) for Total Sleep Time, Sleep onset Latency, Sleep Efficiency, Wake After Sleep Onset, and poor performance with high accuracy and low precision between scorers and automatic sleep scoring (low bias and high s.d.) for REM latency and REM fraction of sleep. Furthermore, we see a systematic shift (high difference in bias) where automatic sleep scoring overestimates N2 fraction of sleep, and underestimates N3 fraction of sleep.

In order to understand how the quality of estimates differed between metrics we plotted bias vs s.d. (Fig 6). The majority of data points are located in the lower left with low variability and bias indicating a good agreement between the estimates. The large deviants are the automatic sleep scorings of REM latency, REM fraction of sleep and discrepancy between N2 and N3 fractions of sleep among the manual scorers. As to the latter, we performed a post hoc analysis and found that most of the disagreement had to do with the timing of the transition from N2 to N3. Regarding REM latency, it is bound to give large variation if an entire REM period was skipped by an estimate. We therefore illustrated how often estimates identified the same period and how often one of the estimates skipped a period. We found that the two manual scorers agree more often than with the automatic algorithm (about REM periods) and that each scorer missed an initial REM period with about equal chance. The automatic algorithm tends to miss the first REM period with a somewhat higher chance of 20%.

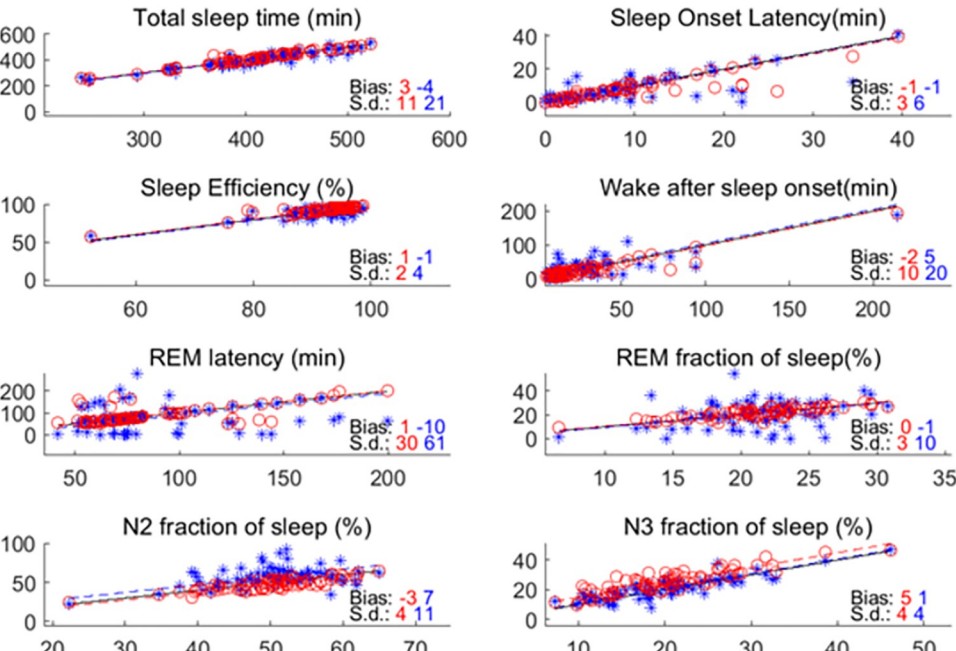

**Fig 5. Eight panels, one for each sleep metric.** The x-axis is the value for scorer1. The y-axis is the alternative score, which is either scorer2 or automatic sleep scoring. The red color indicates comparison between scorer1 and scorer2 (o). The blue color indicates comparison between scorer1 and automatic sleep scoring (*). 's.d.' refers to the standard deviation of the differences between S1 and the alternative scoring. 'Bias' refers to the average difference.' Solid line is the identity. Dashed line is the identity+bias.

To evaluate whether automatic sleep scoring based on repeated ear-EEG is superior to a single night of manually scored PSG, we compared the uncertainty of the two methods as a function of nights included in the model (Fig 7). Automatic ear-EEG scoring and manual PSG scoring show similar results after one night with the latter being a little bit better. However, already when the ear-EEG is used for two nights and scored automatically it performs better than the manually scored PSG with respect to six of the eight sleep metrics. Only for REM fraction of sleep and N2 fraction of sleep are four nights not enough to outperform one night of PSG.

## Discussion

Clinical practice is gradually changing from only doing a single night of PSG to also include several nights of diagnosing and monitoring with unobtrusive devices. This paper confirms several positive aspects of this transition, as has previously been anticipated in other studies [18].

Sleep scoring is traditionally based on systematic scoring schemes [6]. This is a Bayesian approach with both absolute thresholds of amplitudes and complex systems for reevaluating already scored epochs based on later findings, and is not an exact process. This is reflected in the fact that sleep experts in general do not achieve agreements above 0.8–0.85 in Cohen's kappa [19]. Therefore, we do not have an unambiguous 'true' scoring that can be used in the evaluation of alternatives to the PSG based scoring. To compensate for this absence of 'ground truth' labels, we have based the assessment of the automatic ear-EEG based scoring on scorings provided by two highly experienced and independent sleep technicians. The two scorers achieved an interrater agreement of 0.83.

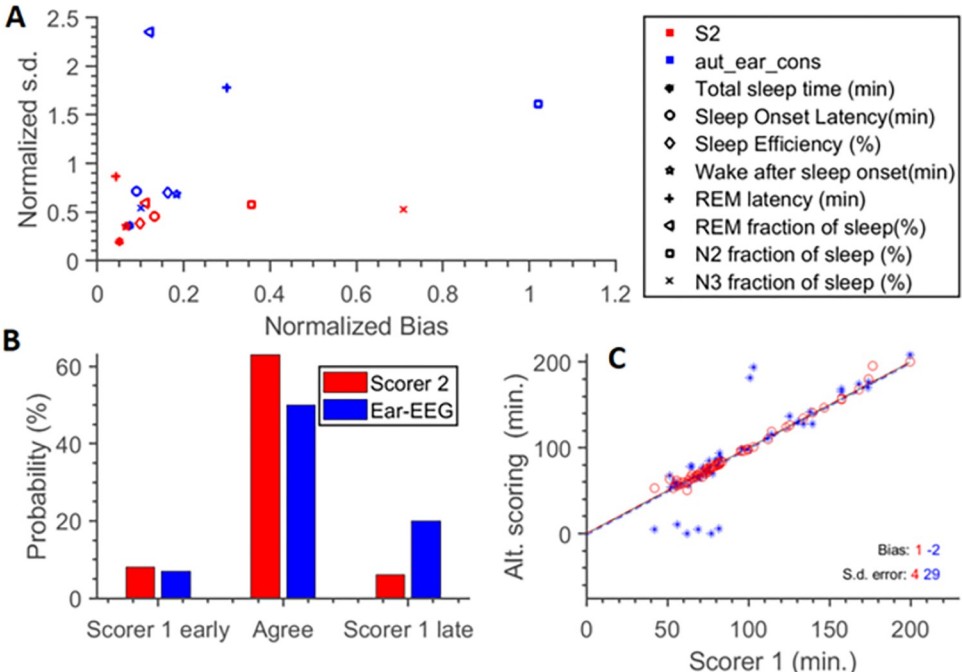

**Fig 6.** Panel A summarizes the main findings in Fig 5 as to how variable the data is (normalized s.d.) plotted against the offset (normalized bias). All bias values are absolute values and normalized to the range 0 to 1. The same normalization factor used on the bias, is used to scale the s.d. The majority of data points are located in the lower left with low variability and bias. The large deviants are the automatic sleep scorings of REM latency and REM fraction of sleep (two top markings) and the discrepancy between N2 and N3 fractions of sleep (two rightmost markings). Panel B toillustrate why the automatic model gives poorer estimates of REM latency than comparing two manual scorers. If the first REM period is missed, then the second REM period will give rise to the latency estimate. If both methods find REM within the first 100 minutes, it is considered as in agreement. It is clear that the two manual scorers agree more often than with the automatic algorithm and that each scorer miss an initial REM period with about equal chance. The automatic algorithm tends to miss the first REM period with a chance of 20%. Panel C identifies the degree of delay in identifying REM-latency. In six cases, the automatic method falsely identifies sleep onset REM.

Prior to evaluating our sleep scoring algorithm with ear-EEG data, we first assessed the performance of the algorithm using PSG data. The algorithm achieved an agreement of 0.80 and 0.81 to scorer 1 and scorer 2 respectively. Thus, only a minor decrease in performance can be attributed to the algorithm. The algorithm based on ear-EEG achieved an agreement in the range from 0.70 to 0.73 (see Fig 3), indicating that the ear-EEG comprises less information about the sleep than a conventional PSG recording. However, the benefit of wearable EEG sleep trackers is that the cost of extra nights is low compared to PSG recordings (precise ratio depending on device used), and a fairer comparison between PSG and ear-EEG (based on healthcare cost and inconvenience for the patient) is to use the ear-EEG for multiple nights. In this study we compare the reliability of sleep metrics calculated from single PSG nights to those calculated from up to four nights. We find that after just a single additional night, the performance cost of using ear-EEG is overcome, and the resulting sleep metric is in fact *more* reliable than the PSG single-night estimate.

This should be seen in the context of our previous work with in-ear-EEG during sleep [8,14,20,21]. The focus has been on sleep in healthy subjects and subjects with epilepsy. To validate ear-EEG as a long-term sleep tracker, we have also performed a 40 nights-study of ear-EEG sleep monitoring during night shifts in a healthcare worker (manuscript under preparation). This approach identified ear-EEG as a useful tool also for long-term sleep monitoring.

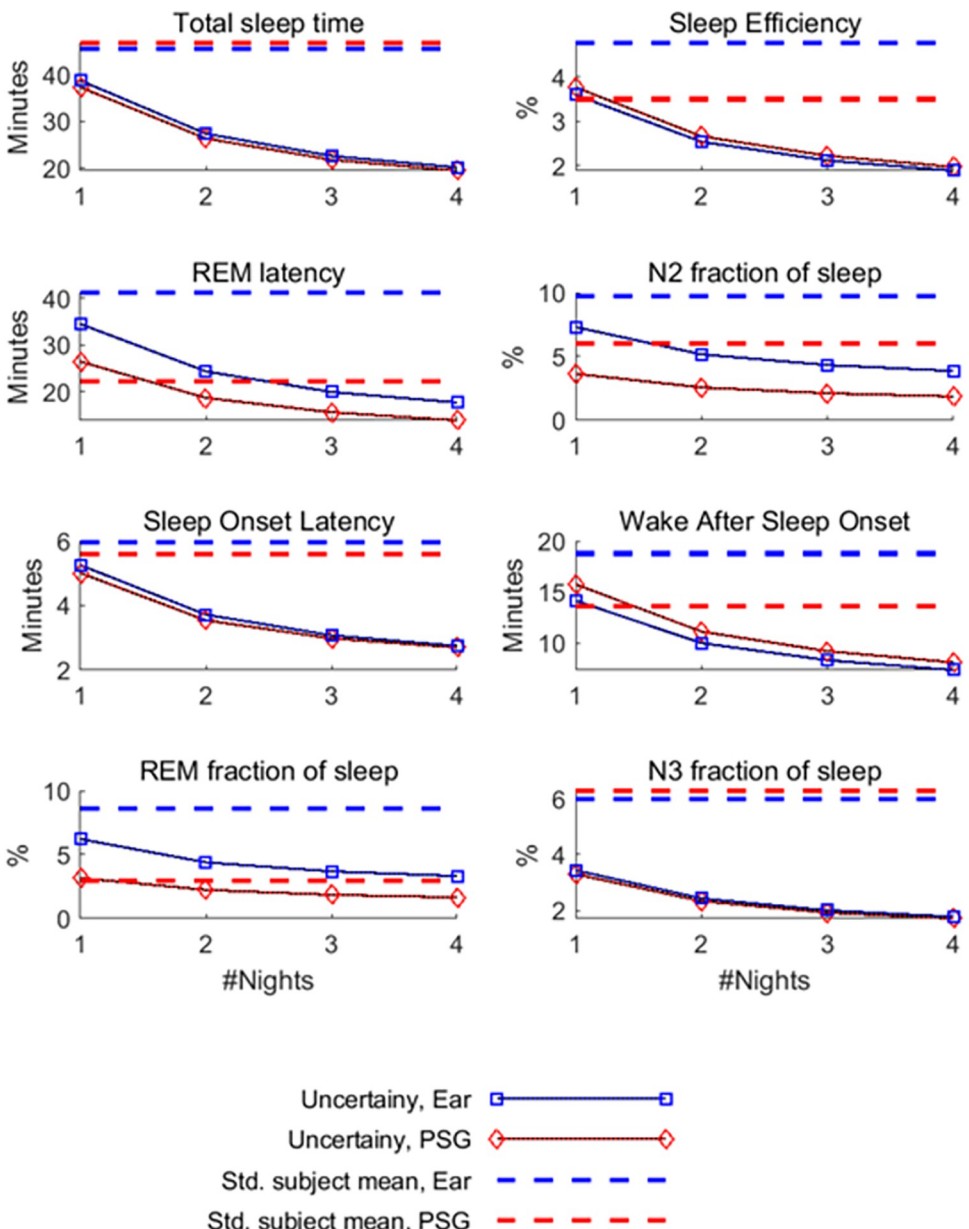

**Fig 7. Benefit of multi-night recordings.** Each panel demonstrates reliability of a given sleep feature as a function of nights recorded. The uncertainty consistently decreases, as more nights are included for each patient (solid lines). The uncertainty for the PSG is smaller than for ear-EEG, but already at the second night, ear-EEG shows lower uncertainty than single-night PSG (for six of the eight sleep metrics). The PSG generally shows less variation than the ear-EEG (3 out of 4 comparisons, dashed lines).

Additionally, we have performed a large number of at-home sleep recordings using only ear-EEG, finding that subjects are able to put on the equipment just as well, without assistance [22]. In this context, it is valuable to also point out that EEG based trackers are generally more accurate than, for instance, accelerometers [23].

While PSG registers all data necessary to perform sleep staging according to AASM, some medical grade devices with fewer modalities such as EEG bands, behind the ear, and implantable EEG-devices have entered the market. All of these are inferior to PSG on a single night of

recording [20,24,25] just as is demonstrated here. However, while we have repeatedly shown (here and in Mikkelsen et al 2019 and 2021 [13,14]) that the single night performance is close to PSG, here we also show that when several nights are recorded, wearable sleep monitors can perform *better* than a single night of PSG. While perhaps not surprising, this is very valuable for the ongoing implementation of wearable sleep monitoring in digital healthcare.

## Limitations

This is a study of healthy young subjects. Such a population is known to have PSGs that are easier to read than the population in general. It is therefore not clear how well our findings generalize to patients. When methods are used for disease diagnosis and classification, it is important that they be validated on relevant populations. Thus, as the findings in this study were based on recording from young and healthy subjects, generalizations to other populations and abnormal sleep should be done with caution.

## Future perspectives

While automatic sleep scoring based on ear-EEG is a promising tool for sleep staging in healthy subjects, it is probably immature to substitute ear-EEG for PSG when clinical diagnosis should be made. Randomized clinical trials and larger cohort studies in patients are needed to verify the positive results presented here. When several days of sleep monitoring is needed, the ear-EEG is a promising, unobtrusive tool.

## Author Contributions

**Conceptualization:** Troels Wesenberg Kjaer.

**Data curation:** Martin Christian Hemmsen.

**Formal analysis:** Mike Lind Rank, Preben Kidmose, Kaare Mikkelsen.

**Investigation:** Troels Wesenberg Kjaer.

**Supervision:** Preben Kidmose.

**Writing – original draft:** Troels Wesenberg Kjaer.

**Writing – review & editing:** Troels Wesenberg Kjaer, Mike Lind Rank, Martin Christian Hemmsen, Preben Kidmose, Kaare Mikkelsen.

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
