## [Decision Letter · Decision Letter 0]

13 May 2022

PDIG-D-21-00026

Repeated automatic sleep scoring based on ear-EEG is a valuable alternative to manually scored polysomnography

PLOS Digital Health

Dear Dr. Kjaer,

Thank you for submitting your manuscript to PLOS Digital Health. After careful consideration, we feel that it has merit but does not fully meet PLOS Digital Health's publication criteria as it currently stands. Therefore, we invite you to submit a revised version of the manuscript that addresses the points raised during the review process.

We look forward to receiving your revised manuscript.

Kind regards,

Henry Horng-Shing Lu

Section Editor

PLOS Digital Health

Journal Requirements:

1. Please amend your Financial Disclosure statement. If you did not receive any funding for this study, please simply state: “The authors received no specific funding for this work.”

2. Please update your Competing Interests statement. If you have no competing interests to declare, please state: “The authors have declared that no competing interests exist.”

3. In the online submission form, you indicated that “Data may be shared after approval by Danish authorities if a collaboration is established”. All PLOS journals now require all data underlying the findings described in their manuscript to be freely available to other researchers, either 1. In a public repository, 2. Within the manuscript itself, or 3. Uploaded as supplementary information.

4. Please provide separate figure files in .tif or .eps format only and remove any figures embedded in your manuscript file. Please also ensure that all files are under our size limit of 10MB.

For more information about how to convert your figure files please see our guidelines: https://journals.plos.org/digitalhealth/s/figures

5. All figures and supporting information files will be published under the Creative Commons Attribution License (creativecommons.org/licenses/by/4.0/). Authors retain ownership of the copyright for their article and are responsible for third-party content used in the article. 

Figure 1A includes an image of an identifiable person. Please provide written confirmation or release forms, signed by the subject(s) (or their parent/legally authorized guardian), giving permission to be photographed and to have their images published under our CC-BY 4.0 license. 

Otherwise, we kindly request that you remove the photograph.

Please upload any written confirmation as an 'Other' file type. It must clarify that the copyright holder understands and agrees to the terms of the CC BY 4.0 license; general permission forms that do not specify permission to publish under the CC BY 4.0 will not be accepted. Note that uploading an email confirmation is acceptable.

Additional Editor Comments (if provided):

Reviewers' comments:

Reviewer's Responses to Questions

**Comments to the Author**

1. Does this manuscript meet PLOS Digital Health’s publication criteria? Is the manuscript technically sound, and do the data support the conclusions? The manuscript must describe methodologically and ethically rigorous research with conclusions that are appropriately drawn based on the data presented.

Reviewer #1: Partly

2. Has the statistical analysis been performed appropriately and rigorously?

Reviewer #1: I don't know

3. Have the authors made all data underlying the findings in their manuscript fully available (please refer to the Data Availability Statement at the start of the manuscript PDF file)?

Reviewer #1: No

4. Is the manuscript presented in an intelligible fashion and written in standard English?

Reviewer #1: Yes

5. Review Comments to the Author

Reviewer #1: GENERAL COMMENTS

Dear authors,

I appreciated the reading of your article.

However, I have several concerns about the study setting, some missing details and the reliability of the results, because, if I correctly understand, the authors compared four nights of the wearable device with one night of PSG.

Please see the following comments

INTRODUCTION

1. Sleep represents an issue also in cancer. I suggest adding the following article: Castelli et al., (2021), DOI: 10.1007/s00520-021-06377-5.

2. In my opinion, respiratory disease instead of respiratory issues is more appropriate.

3. Please pay attention that the cited articles are not correctly reported in the text. Furthermore, they are reported in different formats. 

4. Please check that all the cited articles are present in the reference list.

5. The authors say only wearable. What does this mean? Which kind of device. Maybe, in the introduction, there should be more information to introduce the reader to the device and its advantages regarding PSG.

6. The aim is not precise; the authors just reported the tested hypothesis. 

MATERIALS AND METHODS

1. Lots of information are missing about participants, and the protocol is not clear: where and how have they been recruited? Which were the inclusion and exclusion criteria? Where did they sleep for the assessments night?

2. Did the participants also use the PSG? Otherwise, how did the authors obtain the PSG data for the comparisons?

3. Where did the evaluation take place?

4. The model, the characteristics, the manufacturer and other important details about the device are not reported.

5. Who elaborated the computer algorithm?

6. How did the authors evaluated and choose the two sleep technologists?

7. In my opinion, the authors should report how many epochs they discarded, because this could have influenced the goodness of the data.

8. Why did the authors decide not to evaluate the N1 fraction of sleep?

9. Statistical analysis is not clear, and it is not easy to follow the exposition of the results without a more precise explanation:

1. What does “j nights” mean?

2. Which statistical software has been used?

3. Is it possible to explain better the confusions matrices?

4. Which is the post-hoc used later in the results section?

5. In general, lots of details are reported below the figures and not in the text; in my opinion, the authors should evaluate to integrate the information below the figures in the text.

RESULTS

1. “ in the range 0.7 to 0.74, indicating a substantial agreement”. Does this result mean that without a sleep technologist, the data could be interpreted correctly?

2. In figure $, I suggest adding A, B, C in each panel to make it more understandable. It is not clear to which panel the authors refer.

3. I am wondering if it is correct comparing one PSG night to 4 wearable nights. Probably this is the intent of the authors and the article, but the wearable device is not precise in evaluating only one night, and it becomes more reliable after four nights. Did the authors test more nights? For example, one week? Why did they focus on four nights?

4. Table 1 is not recalled in the text.

DISCUSSION

1. The authors should try to write the discussion more fluently; it appears as many paragraphs not connected one each other.

2. “We have worked extensively…” I do not understand the meaning of this paragraph.

3. The authors should also consider the insertion of some consideration about the use of actigraph.

4. Are these first results or first evidence or what else referring to this device? Are these data really usable from other research groups since the sample size is limited, in a healthy status and with a restricted age?

5. Maybe, the authors should describe a way for making this article more indispensable or useful for future studies.

6. PLOS authors have the option to publish the peer review history of their article (what does this mean?). If published, this will include your full peer review and any attached files.

**Do you want your identity to be public for this peer review?** For information about this choice, including consent withdrawal, please see our Privacy Policy.

Reviewer #1: Yes: Lucia Castelli

---

## [Decision Letter · Decision Letter 1]

31 Aug 2022

PDIG-D-21-00026R1

Repeated automatic sleep scoring based on ear-EEG is a valuable alternative to manually scored polysomnography

PLOS Digital Health

Dear Dr. Mikkelsen,

Thank you for submitting your manuscript to PLOS Digital Health. After careful consideration, we feel that it has merit but does not fully meet PLOS Digital Health's publication criteria as it currently stands. Therefore, we invite you to submit a revised version of the manuscript that addresses the points raised during the review process.

Please submit your revised manuscript within 30 days Oct 30 2022 11:59PM. If you will need more time than this to complete your revisions, please reply to this message or contact the journal office at digitalhealth@plos.org. Please include the following items when submitting your revised manuscript:

We look forward to receiving your revised manuscript.

Kind regards,

Henry Horng-Shing Lu

Section Editor

PLOS Digital Health

Journal Requirements:

Additional Editor Comments (if provided):

Reviewers' comments:

Reviewer's Responses to Questions

**Comments to the Author**

1. If the authors have adequately addressed your comments raised in a previous round of review and you feel that this manuscript is now acceptable for publication, you may indicate that here to bypass the “Comments to the Author” section, enter your conflict of interest statement in the “Confidential to Editor” section, and submit your "Accept" recommendation.

Reviewer #1: All comments have been addressed

2. Does this manuscript meet PLOS Digital Health’s publication criteria? Is the manuscript technically sound, and do the data support the conclusions? The manuscript must describe methodologically and ethically rigorous research with conclusions that are appropriately drawn based on the data presented.

Reviewer #1: Yes

3. Has the statistical analysis been performed appropriately and rigorously?

Reviewer #1: Yes

4. Have the authors made all data underlying the findings in their manuscript fully available (please refer to the Data Availability Statement at the start of the manuscript PDF file)?

Reviewer #1: Yes

5. Is the manuscript presented in an intelligible fashion and written in standard English?

Reviewer #1: Yes

6. Review Comments to the Author

Reviewer #1: Dear authors,

The manuscript significantly improved after the revision process and is now more understandable.

I still have some minor comments:

1. Please write the AASM abbreviation in full.

2. In my opinion, you should write why you decided not to evaluate N1.

3. In my opinion, the fact that the automatic and the manual soring could be considered superimposable should be highlighted in the manuscript. Moreover, does this result means that ear-EEG recording could be performed at home or is the hospitalization still necessary?

7. PLOS authors have the option to publish the peer review history of their article (what does this mean?). If published, this will include your full peer review and any attached files.

**Do you want your identity to be public for this peer review?** For information about this choice, including consent withdrawal, please see our Privacy Policy.

Reviewer #1: No

---

## [Decision Letter · Decision Letter 2]

25 Sep 2022

Repeated automatic sleep scoring based on ear-EEG is a valuable alternative to manually scored polysomnography

PDIG-D-21-00026R2

Dear dr Mikkelsen,

We are pleased to inform you that your manuscript 'Repeated automatic sleep scoring based on ear-EEG is a valuable alternative to manually scored polysomnography' has been provisionally accepted for publication in PLOS Digital Health.

Best regards,

Henry Horng-Shing Lu

Section Editor

PLOS Digital Health

Reviewer Comments (if any, and for reference):

Reviewer's Responses to Questions

**Comments to the Author**

1. If the authors have adequately addressed your comments raised in a previous round of review and you feel that this manuscript is now acceptable for publication, you may indicate that here to bypass the “Comments to the Author” section, enter your conflict of interest statement in the “Confidential to Editor” section, and submit your "Accept" recommendation.

Reviewer #1: All comments have been addressed

2. Does this manuscript meet PLOS Digital Health’s publication criteria? Is the manuscript technically sound, and do the data support the conclusions? The manuscript must describe methodologically and ethically rigorous research with conclusions that are appropriately drawn based on the data presented.

Reviewer #1: Yes

3. Has the statistical analysis been performed appropriately and rigorously?

Reviewer #1: Yes

4. Have the authors made all data underlying the findings in their manuscript fully available (please refer to the Data Availability Statement at the start of the manuscript PDF file)?

Reviewer #1: Yes

5. Is the manuscript presented in an intelligible fashion and written in standard English?

Reviewer #1: Yes

6. Review Comments to the Author

Reviewer #1: (No Response)

7. PLOS authors have the option to publish the peer review history of their article (what does this mean?). If published, this will include your full peer review and any attached files.

**Do you want your identity to be public for this peer review?** For information about this choice, including consent withdrawal, please see our Privacy Policy.

Reviewer #1: **Yes: **Lucia Castelli
